# Structures of highly flexible intracellular domain of human α7 nicotinic acetylcholine receptor

Vasyl Bondarenko [1,9], Marta M. Wells [1,9], Qiang Chen [1], Tommy S. Tillman[1], Kevin Singewald [2], Matthew J. Lawless[2], Joel Caporoso[1], Nicole Brandon [1], Jonathan A. Coleman [3], Sunil Saxena [2], Erik Lindahl[4,5], Yan Xu [1,3,6,7] & Pei Tang [1,6,8✉]

The intracellular domain (ICD) of Cys-loop receptors mediates diverse functions. To date, no structure of a full-length ICD is available due to challenges stemming from its dynamic nature. Here, combining nuclear magnetic resonance (NMR) and electron spin resonance experiments with Rosetta computations, we determine full-length ICD structures of the human α7 nicotinic acetylcholine receptor in a resting state. We show that ~57% of the ICD residues are in highly flexible regions, primarily in a large loop (loop L) with the most mobile segment spanning ~50 Å from the central channel axis. Loop L is anchored onto the MA helix and virtually forms two smaller loops, thereby increasing its stability. Previously known motifs for cytoplasmic binding, regulation, and signaling are found in both the helices and disordered flexible regions, supporting the essential role of the ICD conformational plasticity in orchestrating a broad range of biological processes.

[1] Depatment of Anesthesiology and Perioperative Medicine, University of Pittsburgh, Pittsburgh, PA 15260, USA. [2] Department of Chemistry, University of Pittsburgh, Pittsburgh, PA 15260, USA. [3] Department of Structural Biology, University of Pittsburgh, Pittsburgh, PA 15260, USA. [4] Department of Biochemistry and Biophysics, Science for Life Laboratory, Stockholm University, Solna, Sweden. [5] Department of Applied Physics, Swedish e-Science Research Center, KTH Royal Institute of Technology, Solna, Sweden. [6] Department of Pharmacology and Chemical Biology, University of Pittsburgh, Pittsburgh, PA 15260, USA. [7] Department of Physics and Astronomy, University of Pittsburgh, Pittsburgh, PA 15260, USA. [8] Department of Computational and Systems Biology, University of Pittsburgh, Pittsburgh, PA 15260, USA. [9] These authors contributed equally: Vasyl Bondarenko, Marta M. Wells. ✉email: ptang@pitt.edu

Eukaryotic pentameric ligand-gated ion channels (pLGICs), often called Cys-loop receptors, share a common tripartite structural architecture. Each receptor consists of an extracellular domain (ECD) containing agonist-binding sites, a transmembrane domain (TMD) enclosing a channel gate, and an intracellular domain (ICD) that refers to the regions connecting the TMD helices TM3 and TM4. Compared to the ECD and TMD, the ICD is involved in much more divergent functions. It plays a critical role in receptors' trafficking, localization, and assembly[1]. The ICD also affects channel conductance and desensitization[2,3]. Furthermore, the ICD mediates receptor-receptor interactions[4] and interactions with intracellular proteins that regulate diverse downstream signaling pathways[5]. It has been recognized as a potential target for drug design[6].

Despite its important functional roles, the ICD of Cys-loop receptors lacks comprehensive structure information. Recent advances in the field of structural biology have considerably increased the number of Cys-loop receptor structures, but none included a complete ICD structure. The structurally characterized Cys-loop receptors were often engineered by replacing the native ICD with shorter peptides to augment structure quality[7]. Even without engineering, the ICD structures could at best be partially determined due to inherent structural flexibility in this domain[8,9]. Thus, neither X-ray crystallography nor cryo-EM are suited for resolving full-length structures of flexible ICDs.

The launch of AlphaFold2[10] and RoseTTAFold[11] has revolutionized predictions of protein 3D structures with excellent accuracy. These deep-learning neural network computing software can predict structures of many proteins even without homologs of known structure. However, the prediction confidence depends heavily on the availability of similar local patterns (in the absence of homologs) or well-defined structural packing. In the case of the ICD of Cys-loop receptors, no homolog of known structures is available, and the domain is partly unstructured rather than efficiently packed with few obvious patterns. These lead to predictions that fall in the "very low" confidence category (https://alphafold.ebi.ac.uk/entry/P36544). Thus, new experimental data are critical to resolve structures of this domain.

In this work, we combine experimental structure restraints from nuclear magnetic resonance (NMR) and electron spin resonance (ESR) spectroscopy with Rosetta computations and determine the full-length ICD structures of the human α7 nicotinic acetylcholine receptor (α7nAChR), a major subtype of neuronal nAChRs in the brain. NMR and ESR spectroscopy are well established for providing structural information on flexible or even disordered proteins[12]. The α7nAChR is involved in a wide range of physiological and pathological processes[13]. It is a potential target for the treatment of addiction, Alzheimer's disease, schizophrenia, inflammation and pain, and other disorders[13,14]. It is noteworthy that the α7nAChR's involvement in various processes is mediated not only by its well-characterized ionotropic function, but also by its increasingly recognized metabotropic function[5]. The ICD plays a central role in networking with protein partners associated with diverse downstream signaling pathways[5]. Our α7nAChR ICD structures will facilitate the understanding of these signaling pathways and the design of strategies to manipulate interaction networks for the advancement of therapeutic treatment.

## Results

**α7nAChR constructs suitable for NMR structural investigation of the ICD.** A full-length human α7nAChR[15,16] is well suited for ESR measurements but is too large to perform structural analysis using solution NMR. Thus, for NMR structural determinations, we designed an α7nAChR construct, named TMD + ICD, that encompasses the TMD and ICD with 264 residues (L209-V472) (Supplementary Fig. 1). Although the ICD is the focus of our investigation, the inclusion of the TMD in the construct facilitates a pentameric assembly of the ICD. Indeed, the TMD + ICD forms not only pentamers (Supplementary Fig. 2), but also functional channels as observed previously with the TMD alone[17]. The TMD + ICD channels can be activated by ivermectin and potentiated by the α7nAChR-specific positive allosteric modulator (PAM) PNU-120596 in a concentration dependent manner (Fig. 1a, b). Furthermore, the TMD + ICD in micelles shows similar binding affinities to PAMs as the full-

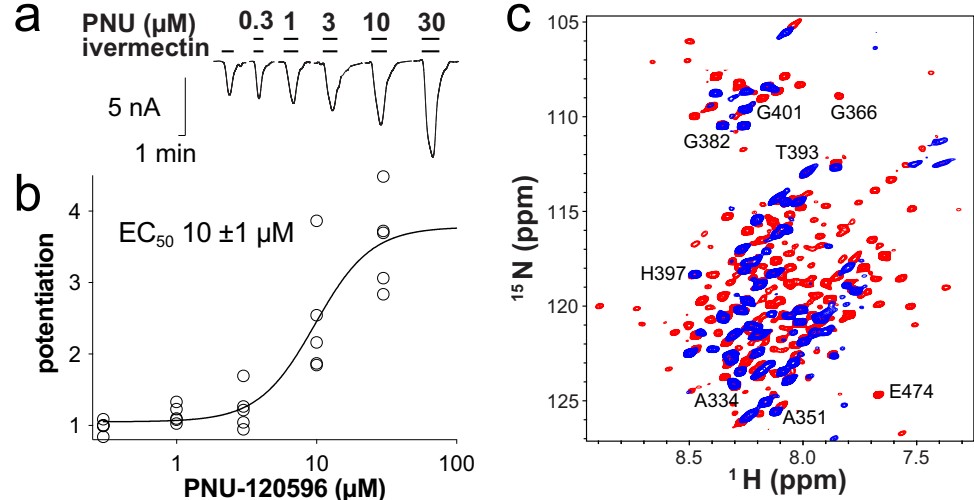

**Fig. 1 Functional α7nAChR TMD + ICD construct and membrane mimetic for structural studies. a** Current traces and **b** concentration-dependent potentiation of *Xenopus* oocytes injected with purified TMD + ICD, showing channel activation by ivermectin (30 μM) and potentiation by PNU-120596. Data shown in black open circles (*n* = 5 sets) were fit to the Hill equation, EC$_{50}$ = 10 ± 1 μM. Source data are provided as a Source Data file. **c** Overlay of $^1$H-$^{15}$N TROSY-HSQC NMR spectra of α7nAChR TMD + ICD in LDAO micelles (red) and in asolectin nanodiscs (blue) collected at an 18.8-tesla NMR spectrometer at 45 °C. Many more residues are observable in micelle samples than in nanodiscs, a trend which is also true for spectra collected at other temperatures (Supplementary Fig. 3).

length α7nAChR (Supplementary Fig. 2). The binding of the intracellular scaffold protein PICK1[16] is virtually the same in the TMD + ICD and the full-length α7nAChR (Supplementary Fig. 2). These data support the functional and pharmacological relevance of the TMD + ICD for structural studies.

The ICD is soluble, but the TMD requires membrane mimetics for stabilization. After extensive screening and testing, we have identified LDAO micelles and asolectin nanodiscs as the mimetics, in which the TMD + ICD is stable and exhibits essentially the same folding, as evidenced by nearly identical NMR chemical shifts ($\Delta\delta < 0.07$ ppm) of the ICD residues (Fig. 1c). However, fewer residues with broader peaks appeared in NMR spectra of nanodisc samples at a range of temperatures (Supplementary Fig. 3) due to the larger size and slower diffusion of nanodiscs in solution NMR. Thus, the micelle mimetic is a better choice for the TMD + ICD based on a combined consideration of protein stability, folding, pharmacological properties, and quality information from NMR spectroscopy.

**TMD+ICD structure determination and a glance at the ICD structures**. The pentameric structures of the TMD + ICD (PDBID: 7RPM) were determined using an iterative protocol (the "Methods" section and Supplementary Fig. 4) with structure restraints derived from NMR and ESR experiments (Supplementary Figs. 5–10; Supplementary Tables 1), which include NMR chemical shifts, nuclear Overhauser effects (NOEs), and residual dipolar couplings (RDCs) for generating secondary structures and Rosetta fragments[18], distance restraints derived from NOEs and paramagnetic relaxation enhancement (PRE) for determining tertiary structures, and quaternary distance restraints derived from the inter-subunit $^{19}$F PREs[19] and $^1$H PREs[20] NMR or from double electron-electron resonance (DEER) ESR[21]. These experimental restraints were integrated into Rosetta[22] to generate an ensemble of pentameric TMD + ICD structures through iterative calculations (Supplementary Fig. 4). The final structures of the TMD + ICD pentamer were validated by an independent set of restraints that had not been used in the structure calculations (see the "Methods" section). The number of various experimental restraints used for structure determination and structure statistics are summarized in Supplementary Tables 1 and 2, respectively. The TMD + ICD structure resembles the available corresponding regions (the TMD and MX/MA helices) in the structures of apo α7nAChR (PDBID: 7EKI)[23] and an antagonist-bound α7nAChR (PDBID: 7KOO)[7] (Supplementary Fig. 11).

The α7nAChR TMD + ICD structure offers a glimpse of a 3D shape of the entire ICD (Fig. 2a–c), which has a height of ~60 Å (measured between Cα atoms of H296 and G402) and a maximum radius of ~50 Å (measured from the Q336 Cα atom to the center of the pore). Each ICD features four regions: a short post-TM3 loop (H296-K307), a MX α-helix (W308-L321) lying nearly parallel to the membrane surface, a large loop L (R322-D408) (Fig. 2d) that could not be determined by cryo-EM[7,23], and a long MA α-helix (P409-C443) that merges with the TM4 helix. A stereo image showing the 15 lowest-energy structures of the TMD + ICD is provided in Supplementary Fig. 12.

Loop L harbors three short helices (h1, P326-K330; h2, K338-R340; and h3, N361-F367) (Fig. 2b). The h3 α-helix in the middle of loop L is anchored onto the MA helix through electrostatic and Van der Waals interactions (R368-E430; F367-R426), dividing loop L into two smaller loops with a "B" shape and thereby increasing the loop stability[24]. Most residues of loop L exhibit faster motion as indicated by their higher NMR signal intensities (peak height) than residues in the MX and MA helices (Fig. 2d). The region containing the h1 α-helix and the h2 $3_{10}$-helix shows

particularly higher mobility as evidenced in the substantially higher NMR intensities, consistent with their positions being most distal from the MA helices (Fig. 2a). The distinctly different dynamics of residues near h3 compared to the rest of the residues in loop L is also evidenced in measurements of backbone dynamics as quantified by NMR $^{15}$N $R_1$ and $R_2$ relaxation rates and $^{15}$N-($^1$H)NOE (Supplementary Fig. 13). Overall, loop L displays substantial disordered regions and flexibility that may be required to accommodate different functions.

**Major interactions stabilizing the ICD**. There is a high fraction of charged residues (18 pairs) in the ICD that distribute unevenly and result in regions with net positive or negative charges (Fig. 3a). The upper region of the ICD has net positive charges, which can interact with the negatively charged inner membrane leaflet. The bottom ICD region is rich with negative-charge residues that potentially bind opposite-charge molecular partners. Charged residues form salt bridges that stabilize the secondary structure (R325-E328; D329-R332; D410-K413), tertiary structure, and quaternary structure. R310-E437 (Fig. 3b) and R368-E430 (Fig. 3c) anchor the MX and h3 helices, respectively, to the MA helix. Salt bridges between adjacent MA helices (E417-R419 and R424-D429) (Fig. 3d) support quaternary structures that are also observed in the partially resolved antagonist-bound α7nAChR[7]. Mutating these residues affected α7nAChR expression[2].

Hydrogen bonding is a dominant force in forming helices, but also exists in forming turns in disordered regions (Fig. 3e) or even supporting quaternary structures (Fig. 3f). These hydrogen bonds have short donor-acceptor distances (<3.2 Å), but their hydrogen atoms often have a donor–acceptor alignment angle of $20° < \theta < 45°$ so that the hydrogen bond strength is weakened compared to that found in the well-structured helices[25]. This type of weak hydrogen bond is better suited for setting dynamic equilibrium among interconvertible conformations in disordered proteins or flexible regions[26].

Despite their lower abundance, hydrophobic interactions exist in disordered loop regions, such as shown in Fig. 3g. Hydrophobic interactions are major contributors to the packing of the bundle of MA helices, especially at the bottom of the ICD.

**Sequence and structure motifs of the ICD**. Motifs relevant to ICD functionality[27] have been found in both structured and disordered regions. Here, using our full-length ICD structure, we attempt to link known sequence and structure motifs to some of their functional engagements. The linkage will help design rigorous studies to further understand the underlying structural bases of functional impacts.

The segment of [307KWTRVILL314] in the first half of the MX helix (Fig. 4) matches the classic mitogen-activated protein kinase (MAPK) docking motif (Kxxxx#x#, where # is a hydrophobic residue), which helps regulate specific interactions in the MAPK cascade[27]. The second half of the MX helix and two residues in the coil (322RM323) makes up the segment of [314LNWCAWFLRM323] (Fig. 4). This segment was found to directly interact with the c-terminus of the N-methyl-D-aspartate receptor (NMDAR) subtype 2 A (NR2A) subunit and participate in the formation of a α7nAChR-NMDAR complex involved in cue-induced reinstatement of nicotine seeking[4]. Upregulation of the α7nAChR-NMDAR complex was observed in brain tissue from rats chronically exposed to nicotine. Furthermore, the α7 peptide [314L-M323] was able to disrupt the complex and block reinstatement of nicotine seeking[4]. A lingering question is whether the peptide [314L-M323] is optimal for blocking reinstatement of nicotine seeking. Are the two coiled residues

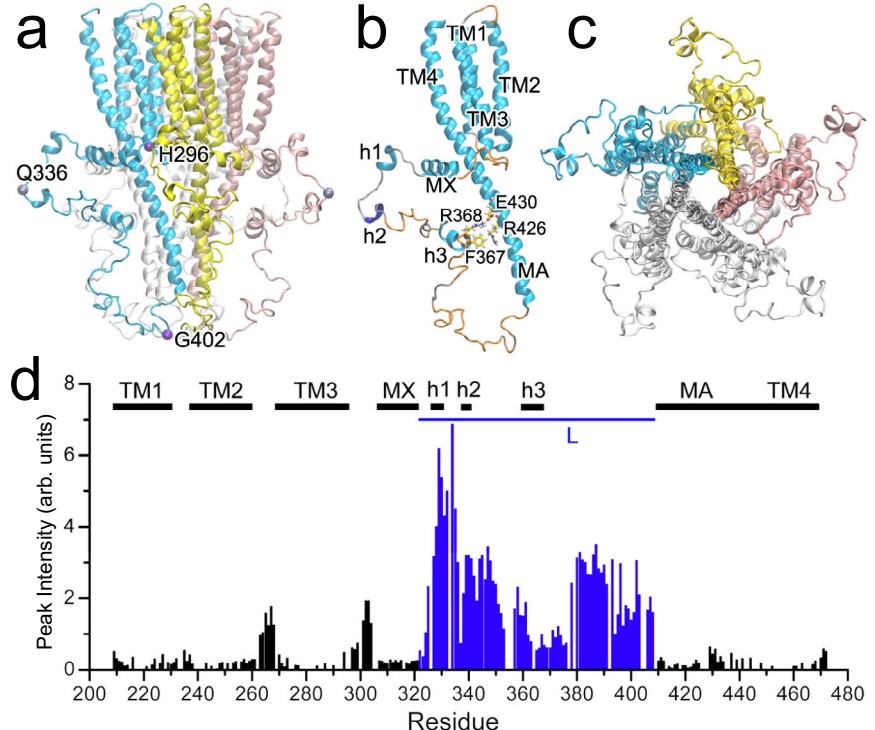

**Fig. 2 Structures of α7nAChR TMD + ICD. a** Side view of a pentameric structure with marked residues (Cα atom shown in VDW) for measuring ICD dimensions. **b** A single subunit color-coded with secondary structures (cyan for α helix, blue for 3₁₀-helix, orange for turn, and silver for coil). **c** A bottom view of the pentameric structure. **d** Intensity (peak height) distribution of TMD + ICD residues obtained from a HNCOCA NMR spectrum. Missing bars are due to overlapping residues or prolines. Note that the distinctly high and low intensities for residues in loop regions and TMD or long ICD helices are indication of faster and slower motions, respectively. Source data are provided as a Source Data file.

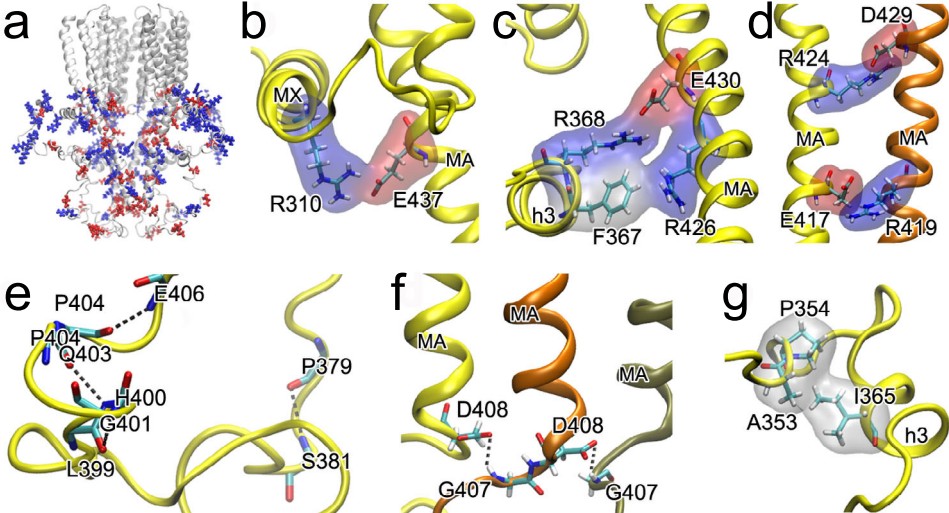

**Fig. 3 Major interactions stabilizing ICD structures. a** Uneven distribution of charge residues (blue: ARG and LYS; red: GLU and ASP) in the ICD. **b–d** Representative electrostatic interactions that stabilize tertiary structure (yellow), and quaternary structure (yellow–orange). **e** Representative hydrogen bonding in disordered loop regions. **f** Representative hydrogen bonding between adjacent subunits. **g** Representative hydrophobic interactions in the ICD. All dash lines in (**c**) and (**d**) < 3.2 Å.

necessary for the peptide function? If yes, can the effect be further improved by adding two more positive charge residues (324KR325) in the coil? These questions remain to be answered by future studies.

The segment of [322RMKR325], identified by mutagenesis, binds Gq proteins to activate downstream signaling pathways involving phospholipase C and inositol triphosphate (IP₃)-mediated intracellular calcium release[5]. Notably, in our ICD structure (Fig. 4), this

disordered segment lies between two α helices (MX and h1). To some extent, this helix-coil-helix structural motif resembles GPCRs' intracellular loops that bind Gq[28]. It will be interesting to determine in the future if the segment of [322RMKR325] selectively binds Gq or if it also binds to other G proteins.

The disordered segment [340RRCSLASVEMS350] (Fig. 4) contains a motif of 340R-346S to mediate protein kinase A (PKA)-type AGC kinase phosphorylation and a 343S-S350 motif

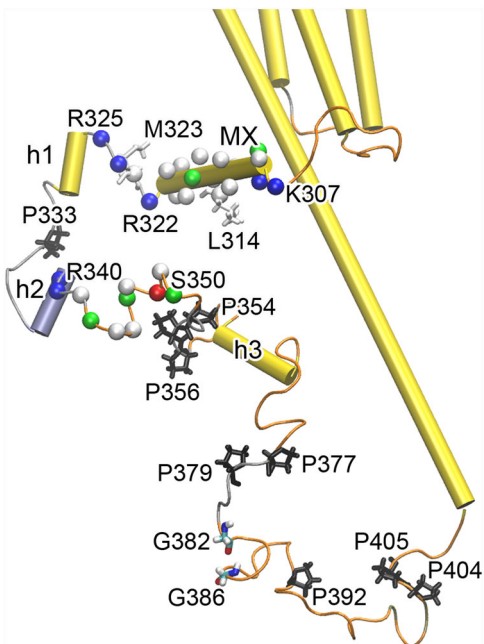

**Fig. 4 Functional relevant motifs in ICD structures.** (**1**) K307-L314, (**2**) L314-M323, (**3**) R322-R325, and (**4**) R340-S350 are represented by CA atom spheres for clarity, with selected residues highlighted with sticks. Residues are colored based on residue type: white-hydrophobic, green-hydrophilic, blue-basic, red-acidic. (**5**) L362-F367, represented by helix h3 in cartoon. (**6**) A GXXXG motif: G382 and G386 forms a hydrogen bond with a distance < 2.1 Å. (**7**) Proline-rich motifs: proline residues are presented in sticks and colored in black.

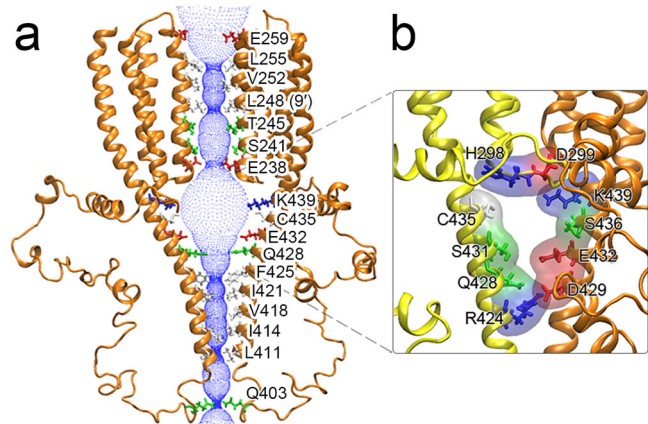

**Fig. 5 Ion permeation pathways. a** Side view of the ion permeation pathway in the α7nAChR TMD + ICD. For clarity, only two subunits are shown. The pore lumen is represented with blue dots. Side chains of pore-lining residues are labeled and shown in sticks with color codes: red or blue for negative and positive charged residues and green or white for hydrophilic and hydrophobic residues, respectively. Source data are provided as a Source Data file. **b** Expanded view of a putative lateral portal formed mostly by charged and polar residues in adjacent MA helices.

to constitute a glycogen synthase kinase-3 (GSK3) phosphorylation site[27]. Note the conformational flexibility of these motifs and their exposure to potential binding partners. The ability to form electrostatic, hydrophobic, and hydrophilic interactions enable these motifs to recruit a diverse set of protein partners.

The segment of [362LLYIGF367] in the h3 helix (Fig. 4) was pinpointed by mutagenesis studies to be essential for the selective transport of α7nAChR to dendrites over axons in hippocampal neurons[29], though the exact mechanism by which these residues regulate α7nAChR targeting was unclear. A tyrosine motif (YXXϕ) in this segment interacts with adaptor protein complexes for transmembrane protein transport and the interaction may facilitate dendritic localization of α7nAChR[30]. The α-helical structure of 362LLYIGF367 benefits the anchoring of the flexible loop to the MA helix (Fig. 2). Whether the α-helical structure is crucial for efficient interactions of the segment with the adaptor protein complexes to regulate α7nAChR targeting requires further investigation.

G382 and G386 form a weak hydrogen bond to fasten a helical turn comprised by a segment of [382GVVCG386] that exhibits a GXXXG motif (Fig. 4). The helical turn can be further stabilized by interactions of local residues (Supplementary Fig. 14). The GXXXG motif has been implicated in mediating transmembrane helix–helix association[31], but its role in the ICD needs to be defined in the future.

Proline-rich motifs occur widely in eukaryotic proteins and serve for various functions in cellular processes[32]. The ICD contains 13 prolines and several proline-rich motifs, including tri-proline (354PPP356), PXP (377PTP379), and XPPX (404PP405) (Fig. 4). Several prolines (P306, P326, P409) at α-helix edges are expected to provide structural flexibility and mobility. Prolines in disordered regions are often exposed for covalent modification and signaling[33].

**Pore conformation and ion permeation pathway.** The pore profile of the α7nAChR TMD + ICD structures shows a non-conductive state (Fig. 5a), with the most constricted radius of ~1.6 Å at L248 (9′) in the TMD that is too small to pass a hydrated Na+ or Ca2+ ion (~3.6 or 4.1 Å radius, respectively). This hydrophobic gate at the 9′ position is conserved in apo[23] and the antagonist-bound[7] nAChRs (Supplementary Fig. 15), as well as in other resting-state cationic pLGICs[8,34,35]. The α7nAChR TMD pore profile resembles that of apo and the antagonist-bound α7nAChR but is distinctly different from those closed-pore profiles of desensitized nAChRs (Supplementary Fig. 15).

The pore widens near the TMD-ICD interface and reaches the maximum vestibule radius of ~9 Å before gradually shrinking into a small hydrophobic girdle, which consists of five hydrophobic residues (F425, I421, V418, I414, L411) along the MA helices with the smallest radius of ~1.5 Å at L411. Note that in the cryo-EM structures of α7nAChR, the small radius of this hydrophobic girdle in an antagonist-bound resting state increased only to a radius of ~3.7 Å in an active state[7], suggesting a low probability for ions permeating through the hydrophobic girdle. Consistent with previous observations in α7nAChR and other homologous channels[7,8], the α7nAChR TMD + ICD presents lateral portals formed by charged and polar residues at the subunit interfaces above the hydrophobic girdle, with a radius of ~3.6 Å (Fig. 5b) that could enlarge to a 4.8 Å radius to allow Na+ or Ca2+ ions to exit from the channel in an active α7nAChR (PDBID: 7KOX)[7]. The relevance of lateral portals to ion permeation has been documented previously by a mutagenesis and electrophysiology study[2], in which mutations of α7nAChR lateral-portal residues caused either a decrease (Q428A, S431A, and E432A) or an increase (H298A and R424A) in channel currents in comparison to WT α7nAChR.

**Discussion**

The α7nAChR TMD + ICD structure captures a complete picture of the ICD. The structured ICD regions resemble the MX and MA helices in the recently reported, partially resolved structures of apo[23] and bungarotoxin-bound[7] resting-state α7nAChRs (Supplementary Fig. 11). The ICD possesses ~57% disordered regions (turns or coils), higher than that in the ECD (~53%) and TMD

(~14%) of α7nAChRs[7,23]. This finding supports the notion that disorder occurs preferentially on the intracellular side[36]. A more important contribution from the current study is the complete structural assessment of loop L in the ICD that could not be resolved by cryo-EM or X-ray crystallography. Our results show that loop L is not simply a structureless coil as predicted by AlphaFold2[10] (https://alphafold.ebi.ac.uk/entry/P36544), albeit in fairness the prediction for this region is flagged as falling in the "very low" confidence category, likely due to the lack of homologs of known structures, few local patterns corresponding to existing structures, and little sidechain packing information from semi-disordered regions. Experimental structures for domains like loop L should provide valuable training data sets to improve deep learning-based predictions for intrinsically disordered proteins or regions in general.

Our NMR data suggest that most residues in loop L experience faster motion and higher flexibility than those of well-structured long helices, as evidenced in their higher NMR peak intensities (Fig. 2d) and negative values of $^{15}$N-($^1$H) heteronuclear NOE (Supplementary Fig. 13) that are regarded as unique among the canonical triad of NMR relaxation parameters and essential for the identification of fast backbone motions. The other two relaxation parameters are longitudinal and transverse relaxation rates, $R_1$ and $R_2$, respectively (Supplementary Fig. 13a, b). $^{15}$N-($^1$H) NOE values become negative when extensive high-frequency (ps-ns) motions are present[37]. The observation agrees with the assertion that intrinsic disorder is often associated with protein flexibility[38]. It is interesting to note distinct changes in $^{15}$N-($^1$H) NOE values along loop L residues, moving from negative to positive near the h3 helix and then negative again (Supplementary Fig. 13c). Such a dynamic pattern matches with the "B"-shape tertiary structure of loop L (Fig. 2b), where the anchoring of h3 helix to the MA helix greatly reduces the frequency of motion of nearby residues and results in positive $^{15}$N-($^1$H) NOE values.

The inherent disorder and flexibility, as observed in the ICD, likely result from evolution. Long disordered regions exist in more than 30% of eukaryotic proteins[39] and 70% of signaling proteins[40]. Disorder is not merely a structural defect. Conformational flexibility of disordered regions is probably required in mediating binding diversity to enable cellular signaling, regulation, and other functions[41,42]. Indeed, without a sufficient conformational flexibility, it is hard to imagine how the α7nAChR ICD could recruit and partner with a diverse group of intracellular proteins for various functions[4,5,27,29,30]. The same assessment is likely applicable to other Cys-loop receptors, whose ICDs may also contain flexible regions as loop L in the α7nAChR TMD + ICD structures, even though ICDs in this family of receptors have relatively low sequence homology (Supplementary Table 3). Future experiments are needed to reveal common and individualized ICD structural properties to guide the understanding of how individual receptors interact with diverse cytoplasmic proteins for different functions.

Despite being largely disordered, the ICD structure presents distinct features. The most noteworthy one is the anchoring of the h3 helix to the MA helix that divides loop L into two smaller loops with a shape of the letter B (Fig. 2). Typically, loop length and stability are inversely correlated[24,43]. Whether a shape of the letter B is a common architecture among ICDs of Cys-loop receptors will be interesting to determine. ICDs of other cation-conducting Cys-loop receptors typically have > 100 or even over 200 residues. Thus, the probability is high for having one anchoring point as occurred in the α7nAChR or even more anchoring points for a longer loop.

The α7nAChR ICD contains plenty of sequence and structure motifs for a wide range of functions (Fig. 4). Each motif may serve multiple functions. Take the α7nAChR MX helix as an example. In addition to its aforementioned functions[4,27], the α7nAChR MX helix may directly interact with RIC-3, a chaperone protein that is known to promote α7nAChR assembly and trafficking[44]. A recent study shows that the mouse 5HT$_{3A}$ receptors' MX helix binds human RIC-3[45]. An unusually high sequence identity (~43%) in MX helices between the human α7nAChR and the mouse 5HT$_{3A}$ receptors (Supplementary Table 3) supports the prediction that the α7nAChR MX helix may also bind RIC-3.

The ICDs of Cys-loop receptors typically have lower sequence similarity, which makes it challenging to identify homologous domains and predict structures. Even within the nAChR family, different subtypes can have a drastically different ICD. For example, α7nAChR has 148 ICD residues but α4nAChR has 266 ICD residues (Supplementary Table 3). Their MX helices share only ~29% sequence identity, which is sufficiently low to differentiate their abilities to bind NMDA receptors and opens opportunities for α7nAChR-targeting therapeutics[4]. Indeed, it has been increasingly recognized[6,46,47] that the divergency in the ICD, compared to the structural conservation of the ECD and TMD, provides greater potential for the development of receptor subtype-selective therapeutics, especially as more and more ICD structural information becomes available.

The success of structure determination of a full-length α7nAChR ICD sheds new light on an area that has been less explored thus far. NMR and ESR spectroscopy, in combination with Rosetta calculations, have provided an effective platform to obtain structural information for the flexible regions of the α7nAChR ICD. The approach used in the current study can be applied to solve ICD structures of other Cys-loop receptors or any structures of intrinsically flexible and disordered regions that cannot be determined by X-ray crystallography and cryo-EM. Notably, the flexible and disordered regions may experience significant conformational changes upon binding to their cellular targets and fold into ordered structures. Thus, the generation of α7nAChR TMD + ICD structures from the current study marks only the beginning of uncovering the mysteries of ICD structural arrangements in the absence and presence of binding partners. Much more is yet to be discovered for the structure-based understanding of signaling pathways mediated by the ICDs that can potentially benefit future structure-based drug discovery efforts targeting specific intracellular signaling.

## Methods

**Protein constructs, expression, and purification**. Full-length human α7nAChR[15] and TMD + ICD were produced in *E. coli* Rosetta 2(DE3) pLysS (Novagen). Expression constructs for the full-length human α7nAChR (UniProtKB P36544: ACHA7_HUMAN) and its cysteine mutants have been reported previously[15,16]. The TMD + ICD (Supplementary Fig. 1) was generated from the TMD construct[17] by adding back the ICD using overlapping PCR and the primers shown in Supplementary Table 4. The constructs containing a single unpaired cysteine were generated as reported previously using the QuickChange Lightning Kit (Agilent Technologies), standard PCR protocols[16], and confirmed by DNA sequencing. To obtain proteins for NMR, ESR, or binding studies, each construct was expressed in LB broth or M9 minimal media if isotopic labeling was required. Cells were grown at 37 °C until reaching an OD$_{600}$ of 0.8 and then cultures were cooled to 15 °C before induction with 0.2 mM isopropyl β-D-thiogalatopyranoside (MilliporeSigma). For full-length α7nAChR, the induction media was supplemented with 0.5 M sorbitol, 10 mM MgCl2, and 10 mM choline. Proteins were expressed at 15 °C for ~16 h or ~72 h in LB or M9 media, respectively, and purified as described previously[15] with the following modifications: cells harvested from 1 liter induction medium were suspended in 150 ml (TMD + ICD) or 600 ml (full-length α7nAChR) lysis buffer (50 mM Tris, pH 8, 500 mM NaCl, and HALT protease) and lysed using a Microfluidics M-110Y microfluidizer. The cell lysate was adjusted to 0.15% (w/v) dodecylphosphocholine (DPC, Anatrace) or 1% (w/v) lauryldimethylamine *N*-oxide (LDAO, MilliporeSigma) with 20 mM imidazole, incubated for 1 h, and then centrifuged at 20,000 × *g* for 20 min. The supernatant was incubated with 2 ml of NiNTA resin (GEHealthcare) for 1 h with gentle mixing. The resin was washed to a flat baseline with buffer at pH 8 containing 100 mM

imidazole, 300 mM NaCl, 0.02 mg/ml asolectin (MilliporeSigma) with either 0.2% DPC or 0.4% LDAO, and the protein was eluted by adjusting the imidazole concentration to 300 mM. The pentamer fraction was isolated by size exclusion chromatography (SEC) using a S200 10/300 column (GEHealthcare) equilibrated with 20 mM HEPES pH 7.4, 300 mM NaCl, 0.2% DPC or 0.1% LDAO. Protein purity was confirmed by SDS-PAGE.

**Electrophysiology measurements**. TEVC measurements of *Xenopus* oocytes injected with purified α7nAChR TMD + ICD (Fig. 1a) were performed as follows: 5 ng of the purified α7nAChR TMD + ICD reconstituted in asolectin liposomes was injected into *Xenopus laevis* oocytes (stages 5–6). After 1–2 days, channel function was measured in a 20 μl oocyte recording chamber (Automate Scientific) perfused at 2.4 ml/min and clamped at −60 mV with an OC-725C Amplifier (Warner Instruments). Recording solutions contained 96 mM NaCl, 2 mM KCl, 1.8 mM CaCl$_2$, 1 mM MgCl$_2$, and 5 mM HEPES at pH 7.0. Data were collected using Clampex 10.6 (Molecular Devices) and processed with Clampfit 10.6 (Molecular Devices). Nonlinear regressions and statistical analysis were performed using Prism software (GraphPad 7.04). The oocytes used in this study were kindly provided by Dr. Thomas Kleyman's lab after harvesting from commercial female *X. laevis* (Xenopus 1, MI). All animal experimental procedures were approved by Institutional Animal Care and Use Committee (IACUC) of University Pittsburgh.

**Surface plasmon resonance**. Steady-state surface plasmon resonance (SPR) responses to α7nAChR positive allosteric modulators (PAMs) (PNU-120596 and TQS) or protein (the PDZ domain of PICK1 binding (Supplementary Fig. 2d, e) were measured at 25 °C using a Biacore 3000 (version 4.1.2) with the NTA sensor chip (GE Healthcare, Uppsala, Sweden). α7nAChR was immobilized to the chip with densities between 1000 and 2000 RU. Responses to PAMs or PICK1 binding were acquired at a flow rate of 30 μl/min using Biacore 3000 acquisition software. After reference and buffer subtraction using BIAEvaluation 4.1.1, apparent dissociation constants were derived by non-linear regression analysis with GraphPad Prism 7.04 using a Langmuir isotherm equation for each PAM or protein-based on three independent injections for each concentration.

**Protein labeling, reconstitution, and sample preparations**. MTSL labeling of unpaired single-cysteine full-length α7nAChR or the TMD + ICD α7nAChR followed the protocol published previously[19,21]. After buffer exchange to phosphate buffered saline (PBS) at pH 8 to remove the reducing agent DTT, a ~15- to 25-fold molar excess of the nitroxide spin label MTSL (Toronto Research Chemicals) was added to protein samples for ~2 h at room temperature, followed by incubation overnight at 4 °C to ensure labeling efficiency > 90%. Free MTSL was removed through dialysis, and then subjected to size exclusion chromatography (SEC) on a Superdex 200 10/300 column (GE Healthcare). For ESR, the purified protein at pH 8.0 was treated with only enough MTSL to achieve the desired 60–80% labeling, and any remaining unreacted MTSL was removed by SEC as described above before confirming the labeling efficiency by ESR.

For RDC NMR experiments, a lanthanide ion (paramagnetic: Tm$^{3+}$ and Dy$^{3+}$; diamagnetic: Lu$^{3+}$) was incorporated into the single-cysteine α7nAChR TMD + ICD constructs using a thiol-specific disulfide reagent, N-[S-(2-pyridylthio)cysteaminyl]-EDTA (Toronto Research Chemicals), following published methods[48]. Briefly, a two-fold molar excess of LnCl$_3$ (Sigma-Aldrich) was added to 10 mM N-[S-(2-pyridylthio)cysteaminyl]-EDTA in 100 mM Tris buffer at pH 7.3 and incubated for 1 h. After incubation, EDTA was added to the solution to adjust the free LnCl$_3$ amount below a 5% excess and the solution was incubated for another 30 min. The TMD + ICD sample was treated with DTT in a 20-fold molar excess for 30 min. After removing DTT, the TMD + ICD sample in 50 mM Tris at pH 8.0, 120 mM NaCl, and 0.5% LDAO was mixed with a 10-fold molar excess of N-[S-(2-pyridylthio)cysteaminyl]-EDTA-Ln$^{3+}$ and incubated for 2 h. All reactions were done at room temperature in the dark. A desalting column was used to remove free small molecules from the protein sample.

The labeled α7nAChR in micelles were used either directly for NMR (Fig. 1c and Supplementary Figs. 3, 6–9, 13) or reconstituted in liposomes or nanodiscs for ESR (Supplementary Figs. 2c, 10) and selected NMR (Fig. 1c; Supplementary Figs. 3a–c) experiments. For protein reconstitution in liposomes, solubilized asolectin was added to purified protein in detergent at a 10-fold weight excess and gently agitated for 1 h before removing detergent using BioBeads. The liposomes were then collected by ultracentrifugation at 200,000 × g for 1 h, and resuspended by sonication (FB11207 FisherScientific) in PBS pH 7.4 prepared in D$_2$O. For the preparation of styrene-maleic-acid (SMA) nanodiscs, the liposomes were solubilized with SMA (Xiran SL30010 P20, Polyscope Polymers BV) by adding 2× the weight of the asolectin present in the liposomes. For the preparation of membrane scaffold protein (MSP)-based nanodiscs, purified protein in detergent was mixed with solubilized asolectin lipids at a molar ratio of 80 lipids to one pentameric protein and mixed thoroughly for 30 min before adding the membrane scaffold protein MSP1D1 to the solution at a molar ratio of 2 (MSP1D1) to 80 (lipids). After incubating for 1 h, detergent was removed using Bio-Beads SM-2 Resin (Bio-Rad Laboratories) or Detergent Removal Resin (Thermo Scientific). A typical NMR sample (Fig. 1c; Supplementary Figs. 3, 6–9, 13) contained 0.2–0.3 mM protein in ~1 % LDAO micelles or in nanodiscs with 5 mM sodium

acetate, pH 4.7, and 25 mM NaCl with 5% D$_2$O for deuterium lock. A typical ESR sample contained 0.1–0.2 mM protein in nanodiscs or liposomes in phosphate-buffered saline pH 7.4 prepared in D$_2$O (Supplementary Figs. 2c, 10).

**NMR spectroscopy for structure and dynamics determination**. NMR spectra were recorded using TopSpin (versions 2.1 and 3.1–3.5, Bruker) on Bruker Avance 700–900 MHz spectrometers equipped with a triple-resonance inverse-detection TCI cryoprobe (Bruker Instruments), processed using NMRPipe[49], and analyzed using OriginPro 8.5 (Origin Lab Corp) and Sparky[50]. In all NMR spectra, the $^1$H chemical shifts were referenced to the DSS resonance at 0 ppm and the $^{15}$N and $^{13}$C chemical shifts were referenced indirectly. A relaxation delay of 1 s was used in NMR data collection unless specified otherwise. The NMR experiments and most relevant acquisition parameters (Supplementary Table 5) are reported. The $^1$H, $^{15}$N, and $^{13}$C chemical shifts were assigned manually based on a suite of 3D spectra, including HNCA and HN(CO)CA with data points of 1024 × 36 × 72 and corresponding spectral windows of 12 × 23 × 28 ppm for the $^1$H, $^{15}$N, and $^{13}$C dimensions, respectively; HNCACB and CBCA(CO)NH (1024 × 36 × 104 points, 12 × 23 × 56 ppm); HNCO (1024 × 56 × 64 points, 12 × 23 × 12 ppm); and $^{15}$N or $^{13}$C-edited NOESY (1024 × 40 × 144 points, 13 × 23 × 13 ppm) with a mixing time of 200 ms (Supplementary Fig. 7). Backbone dihedral angle restraints and order parameters (Supplementary Fig. 5) were derived based on chemical shift of H$^N$, H$^α$, C$^α$, C$^β$, N using TALOS+[51]. $^1$H-$^{15}$N TROSY-HSQC spectra were acquired at different temperatures (Supplementary Fig. 3) (25, 30, 35, 40, and 45 °C) to generate hydrogen-bonding restraints based on temperature coefficients of the backbone amide proton chemical shifts. Residues with coefficients <4.5 ppb/K were considered to have hydrogen bonding in helical regions[52]. To determine RDC NMR restraints, $^1$H-$^{15}$N in-phase/anti-phase (IPAP) TROSY-HSQC spectra were recorded for the single-cysteine TMD + ICD constructs labeled with the paramagnetic N-[S-(2-pyridylthio)cysteaminyl]-EDTA-Tm$^{3+}$/-Dy$^{3+}$ or labeled with the diamagnetic N-[S-(2-pyridylthio)cysteaminyl]-EDTA-Lu$^{3+}$. RDCs for individual resonances were obtained from the difference between the $^{15}$N chemical shifts in the in-phase and anti-phase spectra (Supplementary Fig. 6). To investigate the backbone dynamics of the TMD + ICD (Supplementary Fig. 13), NMR spectra for measuring $^{15}$N spin-lattice (R$_1$) and spin-spin (R$_2$) relaxation rates as well as $^{15}$N-($^1$H) heteronuclear Overhauser effects (hetNOEs) were collected. The spectra for R$_1$ and R$_2$ were acquired with a recycle delay of 3 s; variable delays (τ) ranging from 16–640 ms for R$_2$ or 20–1500 ms for R$_1$. $^{15}$N R$_1$ and R$_2$ relaxation rates and their uncertainties were obtained from the exponential fitting of NMR peak intensities versus the variable delays. The hetNOE spectra were collected with a recycle delay of 5 s. The hetNOE values were calculated as the ratios of peak intensities with and without proton saturation and the associated uncertainties were determined from the signal-to-noise ratio[53]. To derive distance restraints from PRE, $^1$H-$^{15}$N TROSY-HSQC spectra were acquired on a group of single-cysteine TMD + ICD constructs labeled with MTSL (Supplementary Figs. 8, 9), and resulting in peak intensities in the paramagnetic (I) and diamagnetic (I$_0$) conditions in the absence and presence of a ~10-fold excess of ascorbic acid. The intensity ratio (I/I$_0$) and diamagnetic peak linewidth were used to determine the paramagnetic enhancement of the transverse relaxation rate R$_2$$^{sp}$, which was used for deriving distances between the MTSL paramagnetic nitroxide and amid proton of individual residues using the Solomon and Bloembergen equation as described previously[54,55].

**ESR spectroscopy**. To measure labeling efficiency, continuous wave (CW) ESR spectra were collected using Xepr 2.6b.176 on a Bruker ElexSys E680 CW/FT X-band spectrometer equipped with a Bruker ER4122 SHQE-W1 high-resolution resonator at 291 ± 1 K. The parameters for each spin-labeled CW spectrum include a center field of 3514 G and sweep width of 150 G, modulation amplitude of 1 G, modulation frequency of 100 kHz, conversion time of 20.48 ms, and 1024 data points. The doubly integrated intensity of the CW spectrum was compared to a calibration point for quantifying the nitroxide concentration and ultimately the labeling efficiency.

Four pulse DEER experiments were performed on either a Bruker ElexSys E680 CW/FT X-band spectrometer equipped with a Bruker EN4118X-MD4 resonator or Bruker ElexSys E580 Q-band CW/FT spectrometer using an ER 5106-QT2 resonator. All DEER samples were prepared at a spin concentration of 60–160 μM with deuterated glycerol (20% v/v) as a cryoprotectant, and flash frozen in 3 mm I.D. or 2 mm I.D. Pyrex capillary for X-band or Q-band, respectively. DEER experiments followed the pulse sequence [(π/2)$_{v1}$ - τ$_1$ - (π)$_{v1}$ - t+dt - (π)$_{v2}$ - τ$_2$ - (π)$_{v1}$ - τ$_2$ – echo][56]. The pump frequency (v2) was set 70 MHz up-field from the observer frequency (v1). The observe (π)$_{v1}$ pulse length and pump (π)$_{v2}$ pulse length, pump pulse step size (dt), and number of data points (n) are optimized for individual samples with τ$_1$ of 400 ns and τ$_2$ being slightly larger than n*dt. The time domain DEER signal was analyzed with DD[57].

**Structure calculations in rosetta**. Structure calculations used the comparative modeling protocol (RosettaCM)[58] in Rosetta 3.7[59] with the talaris2014 energy function[60]. The resource of Open Science Grid[61] was used for the Rosetta structural calculations. Four major components (Supplementary Fig. 4) are included in iterative calculations. Component 1 (Supplementary Fig. 4a) integrated various experimental restraints (Supplementary Table 1) into RosettaCM, for which

fragment libraries were generated using CS-Rosetta[18] on the Robetta server[62] with input chemical shifts, RDC, and NOE data. Other structure restraints (hydrogen bonds, NOE and PRE, and DEER distance restraints) and the five-fold symmetry were used to restrict the conformational space and improve protein folding. Two template structures from 5-HT₃ receptors (PDBID: 6BE1 and 4PIR)[8,35] were only used in the first round of iterative calculations. Component 2 utilized the input from Component 1 to perform an iterative folding protocol (Supplementary Fig. 4b), which guided conformational sampling towards the global energy minimum while maintaining structural diversity to ensure that calculations did not become trapped in local energy minima. 1,000 structures were generated in each iteration of folding (ICD or later TMD + ICD) with the experimental restraints specified by our experiments. Output structures from each round were ranked by a total score $S_{total}$, a sum of the standard weighted physics-based ($S_{physics}$) and knowledge-based ($S_{knowledge}$) potentials from the talaris2014 energy function[60], and the harmonic ($S_{DEER}$) and sigmoid ($S_{PRE}$) restraint potentials. The top 100 structures were clustered with a 3 Å RMSD cutoff[63] using Matlab 2020b (Mathworks) and the top ranked structures from each cluster were input as new template structures for the next iteration of folding calculations. Folding of the ICD and the TMD + ICD converged after 6 and 16 rounds, respectively (Supplementary Fig. 4e, f). Component 3 was for refinement of the TMD + ICD structures (Supplementary Fig. 4c) that began with Rosetta FastRelax[64], consisting of several cycles of packing and all-atom minimization. The top 50 structures from the last cycle were selected for discrete MD simulations in Chiron[65] to minimize steric clashes, followed by geometry optimization using Phenix 1.19[66]. Finally, in Component 4 (Supplementary Fig. 4d), the structures were validated by the Q-factor[67] (analogous to the crystallographic R-factor) as defined in Eq. 1 using Matlab 2020b (Mathworks):

$$Q = \sqrt{\frac{\sum_i \left(r_{exp}(i) - r_{calc}(i)\right)^2}{\sum_i r_{exp}(i)^2}} \tag{1}$$

where $r_{exp}$ is the experimental restraint distance and $r_{calc}$ is the distance observed in the calculated structure. Q-factors evaluated agreement between calculated and experimental distances and were calculated separately for ESR DEER restraints ($Q^{DEER}$) and for NMR NOE ($Q^{NOE}$) and PRE ($Q^{PRE}$) (Supplementary Table 2). In addition, $Q^{PRE}_{free}$ was calculated using 56 PRE restraints (~10% of total PRE restraints) that were excluded in structure calculations. Other parameters characterizing the quality of the final structures were evaluated using MolProbity[68] and Phenix[66] 1.19 (Supplementary Table 2). VMD[69] was used for structure rendering, visualization, and analysis. Pore profiles were calculated using the HOLE program[70].

**Reporting summary**. Further information on research design is available in the Nature Research Reporting Summary linked to this article.

## Data availability

The atomic coordinates and structural restraints for 15 structures of the apo α7nAChR TMD + ICD have been deposited in the Protein Data Bank with the accession code 7RPM. The chemical shift values have been deposited in the Biological Magnetic Resonance Data Bank (BMRB), accession number BMR30939. The source data underlying Figs. 1, 2, 5 and Supplementary Figs. 2, 4, 5, 8, 10, 13, 15 are provided as a Source Data file. Other data that support the findings of this study are available upon reasonable request to the corresponding author. Source data are provided with this paper.

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

## Acknowledgements
The authors thank Drs. Özge Yoluk and Frank DiMaio for their help in initial Rosetta structure calculations, Dr. Monica Kinde, Mr. Michel Guerrero Tenorio, and Dr. Rieko Ishima for their help in sample preparation and characterizations. Mass spectrometry analysis used the University of Pittsburgh Cancer Institute Cancer Biomarkers Facility that is supported in part by NIH grant P30CA047904. The structural calculations were performed using computational resources provided by the Open Science Grid, which is supported by the National Science Foundation award 1148698, and the U.S. Department of Energy's Office of Science. The reported research was supported by the National Institute on Drug Abuse (NIDA) of the US National Institutes of Health (NIH) under grant Awards Number R01DA046939 (to P.T.). M.M.W. and J.C. were supported by NIH training grants T32EB009403 and T32GM075770, respectively.

## Author contributions
V.B. performed NMR experiments; V.B., M.M.W., Q.C., J.C., and N.B. analyzed NMR data. V.B., M.M.W., and Q.C. performed structure calculations. T.S.T. made protein constructs and performed functional and pharmacological characterizations of the constructs; K.S. and M.J.L. performed ESR experiments and data analysis with the advice from S.S. J.A.C. performed negative-stain imaging. P.T. along with Y.X. and E.L. initiated and designed the project. P.T. wrote the manuscript with input from E.L., Y.X., and other authors. All authors reviewed the results and approved the final version of the manuscript.

## Competing interests
The authors declare no competing interests.
