## [Peer Review File · Nature Communications]

Structures of Highly Flexible Intracellular Domain of Human $\alpha 7$ Nicotinic Acetylcholine ReceptorREVIEWER COMMENTS

Reviewer #1 (Remarks to the Author):

Cys-loop receptors or pentameric ligand-gated ion channels have a domain organization that can be divided as follows: an extracellular ligand binding domain (ECD), a pore-forming transmembrane domain (TMD) and an intracellular domain (ICD). In this study, Bondarenko and colleagues focused their attention to the ICD, which has been elusive in most structural studies so far. This is mostly due to the fact that a large portion of the ICD is intrinsically disordered and cannot be visualized with X-ray crystallography or cryo-EM. Here, Bondarenko used an elegant combination of complementary methods to structurally characterize the TMD+ICD of the human $\alpha 7$ nAChR, which is one of the most abundant subtypes in the human brain. The combination of methods includes nuclear magnetic resonance (NMR) spectroscopy, electron spin resonance (ESR) spectroscopy and Rosetta modeling. Therefore, this is the first time that a study reports the full-length structure of the ICD. This is highly relevant because the ICD plays a central role in networking with protein partners associated with downstream signaling pathways. This information has future implications in disease states involving the $\alpha 7$ nAChR, namely treatment of addiction, Alzheimer's disease, schizophrenia, inflammation, pain among others.

The structure of the full-length ICD largely confirms a partial structure of the ICD that was recently resolved in the cryo-EM structure of the alpha-bungarotoxin resting state of the $\alpha 7$ nAChR published by Ryan Hibbs's lab. The full-length ICD structure reveals that more than 50% of the ICD residues are in flexible regions, with primarily a large "L" loop that has a mobile segment spanning 50 Angstrom from the central channel axis. An alpha-helix anchors loop L onto the MA helix through electrostatic and Van der Waals interactions that divide loop L into two smaller loops, thereby increasing the loop stability. These results pave the way for further studies investigating structural rearrangements of the ICD in the absence and presence of binding partners, thereby opening possibilities for structure-based drug design methods aimed at certain intracellular signaling pathways.

I believe this study will be of interest to a wide audience of ion channel experts and I am supportive of the manuscript as such. I have no major remarks or suggestions.

Reviewer #2 (Remarks to the Author):

The $\alpha 7$ nicotinic receptor functions as a pentameric ligand-gated ion channel and plays diverse signalling roles in human physiology. Recently, the cryo-EM structures of human $\alpha 7$ nicotinic receptor in different states have been published by two independent research groups. In both two papers, a part of ICD, which contains the segments linking helices MX and MA, is invisible in the cryoEM structures. The ICD is

involved in divergent functions, such as making interactions with some important intracellular proteins. Due to the flexibility of this region, neither crystallography nor cryoEM are the suitable tools to obtain its structures. This study solved the complete ICD structures of human $\alpha 7$ using NMR and EPR experiments. It is really a hard work. The ICD structures are complementary to the cryoEM structures of $\alpha 7$. Overall, this study is an important work and provide valuable information about the motifs in ICD for cytoplasmic interaction and regulation.

Minor comments,

1. The authors use E.coli system to express the full-length human $\alpha 7$ nicotinic receptor in this study, but without any figures for protein quality are provided. The authors must provide size exclusion chromatogram and SDS-PAGE results of both full-length $\alpha 7$ and TMD+ICD. Also, the proteins should be identified by mass spectrometry.

2. In this study, many segments in ICD are proposed to engage in the interaction with other proteins, such as NMDAR, G protein, PKA, and so on. Mutagenesis experiments are necessary to validate the structures of ICD and those interactions.

3. The authors compares the pore of TMD+ICD structures with the antagonist-bound $\alpha 7$ cryoEM structure. However, the TMD+ICD could not bind antagonists and there are no antagonists included in the NMR sample. Instead, structural comparison with apo-form $\alpha 7$ (7EKI) is more appropriate.

Reviewer #4 (Remarks to the Author):

In this paper, Bondarenko et al used a combination of solution NMR, ESR, and Rosetta modeling to characterize the structure of the intracellular domain (ICD) of the $\alpha 7$ nicotinic acetylcholine receptor connected to the receptor transmembrane domain (TMD). The functional relevance of the TMD+ICD construct in LDAO micelles has been validated by the expected binding affinity of the positive allosteric modulator PAM. This paper should be a very significant contribution to the field of nicotinic acetylcholine receptor because the highly dynamic ICD is simply not accessible by cryo-EM or X-ray. NMR in combination with molecular modeling constitute the only effective means of providing a glance at this enigmatic region of the receptor. The overall strategy for NMR-based structure determination is sound and effective. But I do have a couple of technical concerns that need to be addressed.

1. Oligomeric state of membrane proteins in detergent micelles has long been a source of controversy. In this study, the NMR sample is in LDAO detergent micelles. The pentameric assembly is indirectly suggested by the DEER ESR profile of non-adjacent/adjacent distance ratio. Can the authors provide independent evidence of pentameric assembly? For example, can you look at the complex by SEC-MALS? Or maybe negative-stain EM can be used to evaluate the homogeneity of the pentamer specie?

2. Maybe I have missed it in the manuscript, but I could not find the number of NMR-derived structural restraints such as intra-subunit and inter-subunit PRE/NOE restraints. They are not listed in Supp Table 2. As you know, probably the most important structural restraints for oligomeric membrane proteins are those of inter-subunit. I can understand that measuring inter-subunit NOEs might be unrealistic for such a large system, but inter-subunit PREs are possible. In Supp Fig. 9, the authors show a very nice strategy to collect inter- and intra- subunit PREs. However, the example illustrated for inter-subunit PRE is not very convincing. At 45 C, I cannot see significant change in peak intensity upon reducing the MTSL. The peak of G302 is too weak to be used to make any conclusions. The authors should show more clearly the important inter-subunit PREs that define the oligomeric arrangement in the ICD.

3. The overlay of the spectra in LDAO and asolectin nanodisc in Supp Fig. 3 makes it more difficult for readers to see the quality of the relevant NMR spectra in the study. The authors should instead only show the spectrum in LDAO and distinguish the TMD and ICD peaks with different colors. It might be also useful to highlight the disordered or flexible regions. The sample is at pH 7.5 and 45 C. Wouldn't the amides of the unstructured loops be not observable due to rapid amide exchange?

4. Despite the decent TROSY-HSQC spectrum at 45 C, backbone assignment is probably still very challenging. The authors need to indicate the fractions of residues unambiguously assigned in the TMD and ICD to allow the readers to evaluate the distribution of NMR data collected for the system. Another detail about the assignment needs clarification. I see a pair of HNCACB and CBCA(CO)NH was recorded. Was CBCA(CO)NH recorded using a protonated or deuterated sample? If protonate sample was used, the relaxation would be much too fast for triple resonance experiments except for the disordered region.

5. I appreciate the authors' effort to measure RDCs for such a large system. My concern, however, is the small magnitude of NH RDCs (~ 2-4 Hz?) used in structure calculation. If the RDCs are in the |2-4| Hz range, one must be extremely careful about characterizing the error of measurement, which is typically 0.3-0.5 Hz for large proteins. For the system under study, I would expect the error to be even larger for weak peaks. In this case, including such small RDCs in the refinement might even introduce more structural noise than improving the structure. I suggest the authors try structural refinement without RDCs. If the authors want to include RDCs, they need to be more thorough in showing RDC distribution and initial fitting of RDCs to the known crystal structure of the TMD to determine an alignment tensor. I don't think these information are provided in the current version of the paper.

In summary, this is a very exciting paper that uses NMR to fill the knowledge gap in a7 nicotinic acetylcholine receptor. Important concerns, however, remain that relates to the validation of pentameric assembly in LDAO as well as inter-protomer NMR restraints. Rigorous data that address these two issues would significantly enhance the overall impact of this work.

We would like to thank all three reviewers for their encouraging comments, constructive critiques, and valuable suggestions. Please find our response to each question below. For clarity, reviewers' comments are in *italics*, our responses are in blue, and new additions to the manuscript are highlighted in red.

Reviewer #1 (Remarks to the Author)

Cys-loop receptors or pentameric ligand-gated ion channels have a domain organization that can be divided as follows: an extracellular ligand binding domain (ECD), a pore-forming transmembrane domain (TMD) and an intracellular domain (ICD). In this study, Bondarenko and colleagues focused their attention to the ICD, which has been elusive in most structural studies so far. This is mostly due to the fact that a large portion of the ICD is intrinsically disordered and cannot be visualized with X-ray crystallography or cryo-EM. Here, Bondarenko used an elegant combination of complementary methods to structurally characterize the TMD+ICD of the human alpha7 nAChR, which is one of the most abundant subtypes in the human brain. The combination of methods includes nuclear magnetic resonance (NMR) spectroscopy, electron spin resonance (ESR) spectroscopy and Rosetta modeling. Therefore, this is the first time that a study reports the full-length structure of the ICD. This is highly relevant because the ICD plays a central role in networking with protein partners associated with downstream signaling pathways. This information has future implications in disease states involving the alpha7 nAChR, namely treatment of addiction, Alzheimer's disease, schizophrenia, inflammation, pain among others.

The structure of the full-length ICD largely confirms a partial structure of the ICD that was recently resolved in the cryo-EM structure of the alpha-bungarotoxin resting state of the alpha7 nAChR published by Ryan Hibbs's lab. The full-length ICD structure reveals that more than 50% of the ICD residues are in flexible regions, with primarily a large "L" loop that has a mobile segment spanning 50 Angstrom from the central channel axis. An alpha-helix anchors loop L onto the MA helix through electrostatic and Van der Waals interactions that divide loop L into two smaller loops, thereby increasing the loop stability. These results pave the way for further studies investigating structural rearrangements of the ICD in the absence and presence of binding partners, thereby opening possibilities for structure-based drug design methods aimed at certain intracellular signaling pathways.

I believe this study will be of interest to a wide audience of ion channel experts and I am supportive of the manuscript as such. I have no major remarks or suggestions.

We appreciate the positive feedback from reviewer #1.

Reviewer #2 (Remarks to the Author)

The $\alpha 7$ nicotinic receptor functions as a pentameric ligand-gated ion channel and plays diverse signalling roles in human physiology. Recently, the cryo-EM structures of human $\alpha 7$ nicotinic receptor in different states have been published by two independent research groups. In both two papers, a part of ICD, which contains the segments linking helices MX and MA, is invisible in the cryoEM structures. The ICD is involved in divergent functions, such as making interactions with some important intracellular proteins. Due to the flexibility of this region, neither crystallography nor cryoEM are the suitable tools to obtain its structures. This study solved the complete ICD structures of human $\alpha 7$ using NMR and EPR experiments. It is really a hard work. The ICD structures are complementary to the cryoEM structures of $\alpha 7$. Overall, this study is an

important work and provide valuable information about the motifs in ICD for cytoplasmic interaction and regulation.

Minor comments,

1. The authors use *E. coli* system to express the full-length human $\alpha 7$ nicotinic receptor in this study, but without any figures for protein quality are provided. The authors must provide size exclusion chromatogram and SDS-PAGE results of both full-length $\alpha 7$ and TMD+ICD. Also, the proteins should be identified by mass spectrometry.

We previously reported (Tillman, *et. al.* 2016, JBC, PMID: 27385587) all details of the *E-coli* expressed the full-length human $\alpha 7$, including confirmation of protein identity by mass spectrometry, protein purity by SDS-page, and oligomerization homogeneity by size exclusion chromatography (Figs 1 and 2, <https://www.ncbi.nlm.nih.gov/pmc/articles/PMC5000075/>)

The results specifically for the $\alpha 7$ TMD+ICD, including the SDS_PAGE and SEC trace, are now added to the Supplementary Fig. 2 to demonstrate the protein quality. The identity of the excised monomer gel band was confirmed to be the TMD+ICD by MS/MS after tryptic digestion at the UPCI Cancer Biomarkers Facility.

Supplementary Fig. 2. The pentameric $\alpha 7$ nAChR TMD+ICD used for NMR structural studies. (a) 15% SDS PAGE gel showing molecular weight markers (lane 1) and purified TMD+ICD (lane 2). Marker molecular weights are shown in kilodaltons. The identity of the monomer band was confirmed to be the TMD+ICD by MS/MS after tryptic digestion at the UPCI Cancer Biomarkers Facility. (b) Size exclusion chromatogram was collected using a Superdex 200 10/300 Increase GL column equilibrated with a buffer containing 10 mM sodium acetate at pH 4.8, 100 mM NaCl, and 0.05% LDAO. The injection point at 0 ml is shown by a vertical pink line. The elution peak at 12.6 ml is consistent with the TMD+ICD pentamer molecular weight. (c) The TMD+ICD construct used for NMR forms pentamers that show a typical non-adjacent/adjacent distance ratio (~ 1.6)² in DEER ESR³ experiments. Data were collected from the TMD+ICD C219 (red) in LDAO micelles and full-length $\alpha 7$ nAChR C435 (black) in liposomes. Both constructs

formed pentamers. **(d)** PNU-120596 and TQS, positive allosteric modulators (PAMs) of $\alpha 7$ nAChR, are known to bind the TMD. In surface plasmon resonance (SPR) measurements, each of the PAMs shows similar binding affinity to the TMD+ICD (black: $K_D = 1.0 \pm 0.2 \mu\text{M}$ for PNU-120596; $8.2 \pm 2.2 \mu\text{M}$ for TQS) and the full-length $\alpha 7$ nAChR (red: $K_D = 1.1 \pm 0.1 \mu\text{M}$ for PNU-120596; $7.8 \pm 3.2 \mu\text{M}$ for TQS), suggesting that the TMD+ICD retains the structural integrity of the full length $\alpha 7$ nAChR. Both the full-length $\alpha 7$ nAChR and TMD+ICD were solubilized in 0.05% LDAO. Note that the measured binding affinities are close to previously reported EC_{50} values ($0.2 \pm 0.06 \mu\text{M}$ for PNU-120596⁴; $6.2 \pm 0.6 \mu\text{M}$ for TQS⁵) measured in different types of cells. **(e)** The intracellular scaffold protein PICK1 PDZ domain binds the TMD+ICD (black: $K_D = 4.2 \pm 0.6 \mu\text{M}$ in DDM; $K_D = 4.0 \pm 0.2 \mu\text{M}$ in LDAO) and the full length $\alpha 7$ nAChR (red: $K_D = 3.3 \pm 0.2 \mu\text{M}$ in DDM; $K_D = 3.1 \pm 0.2 \mu\text{M}$ or in LDAO) with similar affinities in SPR measurements. 0.05% LDAO or 0.05% DDM makes no significant difference to the binding results. $n=3$ for all SPR measurements.

2. In this study, many segments in ICD are proposed to engage in the interaction with other proteins, such as NMDAR, G protein, PKA, and so on. Mutagenesis experiments are necessary to validate the structures of ICD and those interactions.

All the interactions mentioned in the manuscript are based on published works, in which mutagenesis and often many other experiments were performed to validate the $\alpha 7$ ICD's segments/residues involved in those interactions. We have provided references for individual interactions. Mention of these interaction sites is not for validating our structures, but rather to help readers link the structure to previous experimental findings.

3. The authors compared the pore of TMD+ICD structures with the antagonist-bound $\alpha 7$ cryoEM structure. However, the TMD+ICD could not bind antagonists and there are no antagonists included in the NMR sample. Instead, structural comparison with apo-form $\alpha 7$ (7EKI) is more appropriate.

We thank this reviewer for bringing the apo-form $\alpha 7$ (7EKI) to our attention. Indeed, the apo $\alpha 7$ structure is much more appropriate for the structural comparison. The related writing in the main text and Supplemental Figures 11 and 14 have been edited/updated to reflect a comparison of our structure with the apo-form $\alpha 7$ (7EKI).

Supplementary Fig. 11. Comparison of the new $\alpha 7$ nAChR TMD+ICD structure (orange) with the Cryo-EM structures of apo $\alpha 7$ nAChR (blue, PDBID: 7EKI)³¹ and α -bungarotoxin-bound resting-state $\alpha 7$ nAChR (cyan, PDBID: 7KOO)³². **(a)** A side view of the aligned TMD and ICD structures. For clarity, only two subunits are shown. **(b)** A bottom view of the aligned MA and MX helices from the three structures. The structure alignment used the regions of TM1-TM3 (L209-Y295), MX helix (W308-R321), and MA-TM4 helix (D410-P469). Pairwise backbone RMSD values (TMD+ICD vs. Cryo-EM structure) were calculated for all regions ($\text{RMSD}_{7\text{EKI}} = 2.66 \text{ \AA}$; $\text{RMSD}_{7\text{KOO}} = 2.47 \text{ \AA}$) and for the helical regions ($\text{RMSD}_{7\text{EKI}} = 2.21 \text{ \AA}$; $\text{RMSD}_{7\text{KOO}} = 1.94 \text{ \AA}$), which included L209-L231 (TM1), G237-I260 (TM2), P269-Y295 (TM3), W308-R321 (MX), and D410-P469 (MA-TM4). Structure alignment and RMSD calculations were performed using VMD³³.

Supplementary Fig. 14. Pore profile comparisons of the $\alpha 7$ nAChR TMD+ICD with other Cys-loop receptors. The averaged pore profile from 15 structures of the $\alpha 7$ nAChR TMD+ICD (black; the gray shadow shows the standard deviation) is compared with the pore profiles of (1) apo $\alpha 7$ nAChR (blue, PDBID: 7EKI)³⁵, α -bungarotoxin-bound $\alpha 7$ nAChR (crimson, PDBID: 7KOO)³¹, α -bungarotoxin-bound $\alpha\beta\gamma\delta$ AChR (orange, PDBID: 6UWZ)³⁶, nicotine-bound $\alpha 3\beta 4$ nAChR (blue dash, PDBID: 6PV7)³⁷, apo-5HT_{3A}R (green, PDBID: 6BE1)⁶, and serotonin-bound 5HT_{3A}R (green dash, PDBID: 6DG7)³⁸. The pore radius is calculated by HOLE³⁹ and plotted as a function of distance along the pore axis. The transmembrane domain (TMD) and intracellular domain (ICD) are separated by a dashed line. The pore profile of the newly determined $\alpha 7$ nAChR TMD+ICD structure matches reasonably well with the pore profiles of the apo $\alpha 7$ nAChR (blue) and α -bungarotoxin-bound $\alpha 7$ nAChR (crimson). The extra length of the pore profile shown around -80 from the new TMD+ICD structure is contributed by residues in loop L (Fig 2) that are missing in previously published structures.

No critique from Reviewer #3.

Reviewer #4 (Remarks to the Author)

In this paper, Bondarenko et al used a combination of solution NMR, ESR, and Rosetta modeling to characterize the structure of the intracellular domain (ICD) of the $\alpha 7$ nicotinic acetylcholine receptor connected to the receptor transmembrane domain (TMD). The functional relevance of the TMD+ICD construct in LDAO micelles has been validated by the expected binding affinity of the positive allosteric modulator PAM. This paper should be a very significant contribution to the field of nicotinic acetylcholine receptor because the highly dynamic ICD is simply not accessible by cryo-EM or X-ray. NMR in combination with molecular modeling constitute the only effective means of providing a glance at this enigmatic region of the receptor. The overall strategy for NMR-based structure determination is sound and effective. But I do have a couple of technical concerns that need to be addressed.

*1. Oligomeric state of membrane proteins in detergent micelles has long been a source of controversy. In this study, the NMR sample is in LDAO detergent micelles. The pentameric assembly is indirectly suggested by the DEER ESR profile of non-adjacent/adjacent distance ratio. Can the authors provide independent evidence of pentameric assembly? For example, can you look at the complex by SEC-MALS? Or maybe negative-stain EM can be used to evaluate the homogeneity of the pentamer specie? We share the same view with this reviewer. At the early stage of $\alpha 7$ nAChR structural studies, we characterized/confirmed pentameric assemblies of the $\alpha 7$ TMD in LDAO micelles using SEC-MALS (Bondarenko, et. al. *Biochim Biophys Acta*, **2014**, PMID: 24384062). A SEC-MALS characterization was also performed on the $\alpha 4\beta 2$ TMD in LDAO micelles and confirmed a pentameric state (Bondarenko, et. al. *Biochim Biophys Acta*, **2012**, PMID: 22361591). An SEC profile of the purified TMD+ICD is now provided (Supplemental Fig 2) to show the homogeneity of the pentameric TMD+ICD. It would be ideal if we could use SEC-MALS again for the TMD+ICD. Unfortunately, SEC-MALS instrument was*

demolished a few years ago and we have no access to a replacement. Thus, we reached out to Dr. Jonathan Coleman, an expert of EM, for negative-stain images to demonstrate that the TMD+ICD in LDAO forms pentamers. Although a high content of LDAO (to mimic a condition of NMR samples) restrains the quality of images, the TMD+ICD particles at or near “top/bottom” views in negative-stain images show the expected doughnut shape, which is consistent with the DEER ESR data (Supplemental Fig 2c) of a predominant pentameric state of the TMD+ICD in LDAO.

2. *Maybe I have missed it in the manuscript, but I could not find the number of NMR-derived structural restraints such as intra-subunit and inter-subunit PRE/NOE restraints. They are not listed in Supp Table 2. As you know, probably the most important structural restraints for oligomeric membrane proteins are those of inter-subunit. I can understand that measuring inter-subunit NOEs might be unrealistic for such a large system, but inter-subunit PREs are possible. In Supp Fig. 9, the authors show a very nice strategy to collect inter- and intra- subunit PREs. However, the example illustrated for inter-subunit PRE is not very convincing. At 45 C, I cannot see significant change in peak intensity upon reducing the MTSL. The peak of G302 is too weak to be used to make any conclusions. The authors should show more clearly the important inter-subunit PREs that define the oligomeric arrangement in the ICD.*

The numbers of NMR-derived and ESR-derived restraints for $\alpha 7nAChR$ structure calculations (number of restraints per subunit) were/are provided in **Supplementary Table 1**.

Inter-subunit structure restraints resulted mainly from DEER ESR and inter-subunit PREs. As suggested, we added additional residues showing inter-subunit PREs to Supplementary Fig. 9 (see next page). Some of these residues show weaker peaks than others because of their intrinsic motion differences. Inter-subunit PRE experiments also present technical constraints. To ensure the observed PREs resulted only from inter-subunits, samples were prepared with 4/5 of the total TMD+ICD labeled only with MTSL and no isotope labeling (and thus NMR invisible) and 1/5 of the total TMD+ICD ^{15}N -labeled. These samples produced much weaker signals because of the reduced ^{15}N content. Even though we significantly extended data collection time, intensities of some peaks still looked weaker than a normal sample.

A negative-stain image of the purified $\alpha 7nAChR$ TMD+ICD (15 $\mu g/ml$) in LDAO micelles. Several discrete top/bottom views of oligomers are highlighted. Scale bar- 20 nm. The grid (Pure C 400 mesh Cu) was glow discharged for 30 s at 25 mA, protein (15 $\mu g/ml$) was added to the grid for 30 s, followed by three successive washings in MilliQ water, and stained with 0.75% uranyl formate. The grid was imaged on a TF20 (200 kV) at a magnification of 150 kx (0.72 \AA /pixel).

Supplementary Fig. 9. (A continuation from the previous page) **Additional representative spectra showing inter-subunit PREs for quaternary structure determination of $\alpha 7nAChR$ TMD+ICD.** (a) Top view showing two residue-groups (magenta, violet) experiencing inter-subunit PRE resulting from the MTSL-labeled V311C (cyan) in two adjacent subunits. (b)-(d) Representative regions of 1H - ^{15}N TROSY-HSQC PRE NMR spectra collected at 45 °C. Residues labeled in bold demonstrate PRE due to the MTSL-labeled V311C. (e) Side view showing residues (magenta) experiencing inter-subunit PRE resulting from the MTSL-labeled C427 in an adjacent subunit. (f)-(h) Representative regions of 1H - ^{15}N TROSY-HSQC PRE NMR spectra collected at 45 °C. Residues labeled in bold demonstrate PRE due to the MTSL-labeled C427.

3. *The overlay of the spectra in LDAO and asolectin nanodisc in Supp Fig. 3 makes it more difficult for readers to see the quality of the relevant NMR spectra in the study. The authors should instead only show the spectrum in LDAO and distinguish the TMD and ICD peaks with different colors. It might be also useful to highlight the disordered or flexible regions. The sample is at pH 7.5 and 45 C. Wouldn't the amides of the unstructured loops be not observable due to rapid amide exchange?*

We took the reviewer's suggestion and added a new figure to Supplementary Fig. 3d (right). Red is for residues in the ICD, black for residues in the TMD, and green is for overlapping residues from the ICD and TMD. The sample was at 45C and pH 4.9 (the protein could not produce good NMR spectra at pH 7.5). Thus, we did not encounter rapid amide exchange.

Supplementary Fig. 3. Comparison of different membrane mimetics and temperatures on the quality of 1H - ^{15}N TROSY-HSQC NMR spectra of $\alpha 7nAChR$ TMD+ICD. Overlay of the NMR spectra acquired in LDAO micelles (red) and asolectin lipid nanodiscs (blue) at (a) 25°C, (b) 35°C, and (c) 45°C. A greater number of NMR peaks were observed for samples prepared in LDAO micelles than those prepared in nanodiscs at three different temperatures. Also, there is a clear trend of an increasing number of observed residues when the data collection temperatures were raised from 25°C to 45°C. For clarity, residues were selectively labeled for different scenarios: residues shown in both micelles and nanodiscs (such as G382, T393 and many others in loop L), residues shown only in micelles at all tested temperatures (such as G366, E474), and residues shown only in micelles and only at 45°C (such as those residues labeled in bold in c that belong to long helices in the TMD or ICD). (d) **A representative 1H - ^{15}N TROSY-HSQC NMR**

spectrum of $\alpha 7n$ AChR TMD+ICD in LDAO micelles collected at 45°C, where NMR peaks from residues in different domains are color coded: black–TMD; red– ICD; green– overlapping peaks from the ICD with the TMD. Note that the TMD residues have generally weaker intensity except those at the N- and C-termini.

4. Despite the decent TROSY-HSQC spectrum at 45 C, backbone assignment is probably still very challenging. The authors need to indicate the fractions of residues unambiguously assigned in the TMD and ICD to allow the readers to evaluate the distribution of NMR data collected for the system. Another detail about the assignment needs clarification. I see a pair of HNCACB and CBCA(CO)NH was recorded. Was CBCA(CO)NH recorded using a protonated or deuterated sample? If protonate sample was used, the relaxation would be much too fast for triple resonance experiments except for the disordered region.

The chemical assignment for the TMD and ICD was deposited to the BMRB site as requested (BMRB entry 30939). The fractions of residues unambiguously assigned in the TMD and ICD are marked in Supplementary Table 1. It would be ideal if we could have sufficient deuterated proteins for those experiments. However, because of the low yield obtained with deuterated protein, we had to use protonated samples for NMR experiments, including CBCA(CO)NH.

5. I appreciate the authors' effort to measure RDCs for such a large system. My concern, however, is the small magnitude of NH RDCs (~ 2 -4 Hz?) used in structure calculation. If the RDCs are in the $|2$ -4| Hz range, one must be extremely careful about characterizing the error of measurement, which is typically 0.3-0.5 Hz for large proteins. For the system under study, I would expect the error to be even larger for weak peaks. In this case, including such small RDCs in the refinement might even introduce more structural noise than improving the structure. I suggest the authors try structural refinement without RDCs. If the authors want to include RDCs, they need to be more thorough in showing RDC distribution and initial fitting of RDCs to the known crystal structure of the TMD to determine an alignment tensor. I don't think these information are provided in the current version of the paper.

We agree with the reviewer that NH RDCs are too small to be used for structural refinements. As shown in Supplementary Fig 4, RDCs were used only in the initial stage of generating fragments in CS-Rosetta, which uses mainly chemical shifts for prediction of fragment structures and also allows inclusion of RDCs and NOEs to aid in fragment structure prediction. RDCs were not included in the ICD/ TMD+ICD folding calculations (RosettaCM) nor in TMD+ICD structural refinements.

In summary, this is a very exciting paper that uses NMR to fill the knowledge gap in $\alpha 7$ nicotinic

acetylcholine receptor. Important concerns, however, remain that relates to the validation of pentameric assembly in LDAO as well as inter-protomer NMR restraints. Rigorous data that address these two issues would significantly enhance the overall impact of this work.

We have provided additional data for pentameric assembly in LDAO and inter-protomer NMR restraints to mitigate the two concerns.

REVIEWERS' COMMENTS

Reviewer #1 (Remarks to the Author):

I am satisfied with this version of the manuscript. I have no further questions.

Reviewer #2 (Remarks to the Author):

I support the acceptance and publication of this manuscript. I only have one minor comment:

In Supplementary Fig.2a, I wonder if the authors identified the “dimer” band by mass spectrometry, since it’s only mentioned in the figure legend that “The identity of the monomer band was confirmed to be the TMD+ICD by MS/MS ...”. Otherwise, it is better to remove the “dimer” label.

Reviewer #4 (Remarks to the Author):

The authors have adequately addressed two of my biggest concerns: the pentameric integrity of the NMR sample and inter-subunit PRE restraints. I believe the paper is now suitable for publication.

RESPONSE TO REFEREES LETTER

We are pleased to learn that all the three reviewers are satisfied with our revised manuscript and support the acceptance and publication of the manuscript.

Reviewer 2 has a minor comment about the labeling of the SDS-PAGE (Supplementary Figure 2a). We have updated the label in the figure based on this reviewer's suggestion.